# Does opting in or out affect the take up of incentives in a long running population-based cohort study: A nested randomised trial in ALSPAC

**Kate Northstone\*, Claire Bowring**

Population Health Sciences, Bristol Medical School, University of Bristol, Bristol, United Kingdom

\* Kate.Northstone@bristol.ac.uk

## Abstract

### Background

Financial incentives may be important for improving response rates to data collection activities and for retaining participants in longitudinal studies. However, for large studies, this introduces significant additional costs. We sought to determine whether an opt-in or an opt-out option for receiving financial incentives when completing questionnaires offers any cost saving measures.

### Methods

The Avon Longitudinal Study of Parents and Children has been ongoing for more than 30 years. It has offered a £10 incentive for returning a partly or fully completed annual questionnaire for >10 years, this is provided by default unless a participant chooses to opt out. For questionnaires completed in 2020 by the original parents recruited to the study and by their offspring, we randomised eligible participants to either opt-out or to opt-in to receiving their vouchers. Logistic regressions determined whether opt-out or opt-in made any difference to the proportion of respondents receiving their vouchers.

### Results

Respondents are less likely to choose to receive a thank you for their time in the form of a £10 shopping voucher if they are asked to opt in compared to if they are asked to opt out. The odds ratio, adjusted for baseline characteristics was 3.94 (95% Confidence Interval: 3.49, 4.45). There was no difference in response rates according to whether respondents were randomised to the opt-in or opt-out group.

### Conclusions

ALSPAC now employs an opt-in procedure for respondents receiving their financial incentive when completing a questionnaire. We recommend similar studies that rely on volunteers

**Data Availability Statement:** The informed consent obtained from ALSPAC participants does not allow the data to be made freely available through any third party maintained public

repository. However, data used for this submission can be made available on request to the ALSPAC Executive. The ALSPAC data management plan describes in detail the policy regarding data sharing, which is through a system of managed open access. Full instructions for applying for data access can be found here: http://www.bristol.ac.uk/alspac/researchers/access/. The ALSPAC study website contains details of all the data that are available (http://www.bristol.ac.uk/alspac/researchers/our-data/).

**Funding:** The UK Medical Research Council and Wellcome (Grant ref: 217065/Z/19/Z) and the University of Bristol provide core support for ALSPAC. The funders had no role in study design, data collection and analysis, decision to publish, or preparation of the manuscript. This publication is the work of the authors and KN will serve as guarantor for the contents of this paper. A comprehensive list of grants funding is available on the ALSPAC website (http://www.bristol.ac.uk/alspac/external/documents/grant-acknowledgements.pdf).

**Competing interests:** The authors have declared that no competing interests exist.

**Abbreviations:** ALEC, ALSPAC Law and Ethics Committee; ALSPAC, Avon Longitudinal Study of Parents and Children; G0, Generation 0 (original parents enrolled in the study); G1, Generation 1 (originally offspring of the parents enrolled in the study).

consider this option if they want to introduce some cost savings without harming overall response rates.

## Background

Effective strategies are critical to engage and retain participants taking part in long term studies. A recent systematic review identified that the most common retention strategy employed was a cash/voucher incentive (59/95 studies reviewed) [1]. However, the authors acknowledged high levels of heterogeneity in this review due to the range of methods, study types and hypotheses examined in the reviewer papers [1]. Nevertheless, there is also evidence to suggest that monetary incentives can improve selection bias in long term research studies [2, 3].

The Avon Longitudinal Study of Parents and Children has been collecting data from its participants via a number of methods for over 30 years. Participants in the study (parents—generation 0 (G0), and their offspring—generation 1 (G1) now aged ~31 years) complete questionnaires annually and they are offered a small incentive of a £10 high street shopping voucher for returning a partly or fully completed a questionnaire. These vouchers can be used in many UK shops and online stores including supermarkets, general household, health and gift stores. This has received full ethical approval from the ALSPAC Law and Ethics Committee (ALEC). Across both generations we would anticipate upwards of 10,000 completed questionnaires. This comes at a substantial financial burden of over £100,000 in incentives alone.

Anecdotally, we are aware that some participants do not use their voucher once it has been issued or that they would be happy to complete their questionnaires regardless of being incentivised. However, we have evidence to suggest that providing an incentive does offer improvement to response rates: We did not offer incentives for our 21-year questionnaire, to G1 respondents, and the response rate dropped by 5.5% compared to the previous questionnaire, completed at 20 years of age. It recovered to previous rates in the next questionnaire (age 22 years) when incentives were offered again. When incentives have been offered in recent years, the study has provided an opt-out option for these respondents to indicate that they do not wish to receive their voucher, however, we were interested to see whether an opt-in option could introduce cost saving, without impacting response rates, given the large cost associated with providing incentives in ALSPAC as noted above. This change in strategy was informed by the "default effect". This theory suggests that individuals are more likely to accept a default option rather than indicate a different course of action [4]. A frequently cited example from Johnson & Goldstein [5] demonstrates this effect very clearly: They showed that using opting in to organ donation resulted in a much lower sign up of organ donors compared to using opting out. Indeed, Jachimowicz *et al* reported in their meta-analysis of 58 studies examining default effects that opt-out defaults led to greater uptakes compared to opt-in [6]. However, this effect was moderated by different characteristics of the study types, with substantial heterogeneity seen in effect sizes according to study type. Nevertheless, we hypothesised that changing the option for receiving a voucher from opt-out to opt-in would *reduce* the uptake of vouchers and thus provide some cost savings.

We therefore randomised eligible participants who received invitations to complete an annual questionnaire sent out to both generations in late 2019/early 2020 to opt-out of receiving their voucher or to opt-in to receiving it.

## Methods

The primary aim of the study was to determine whether an opt-in or an opt-out option for receiving financial incentives for returning a partly or fully completed questionnaire offers any

cost saving measures for the study. Our secondary aim was to determine the characteristics of those respondents who chose to opt-out versus those who chose not to opt-in and to see whether response rate was affected according to whether respondents were in the opt-in or opt-out arm.

## Study population

The Avon Longitudinal Study of Parents and Children (ALSPAC) is an ongoing population-based, birth cohort based in Bristol in Southwest England. Pregnant women resident in the area with expected dates of delivery between 1st April 1991 and 31st December 1992 were invited to take part. These women, their partners (collectively known as G0) and their resulting children (known as G1) [7, 8] have been followed up ever since. 14,541 pregnancies were originally enrolled, however, when the oldest children were approximately 7 years of age, an attempt was made to bolster the initial sample with eligible cases who had failed to join the study originally. This resulted in an additional 913 children being enrolled. 14,901 children were alive at 1 year of age [9] and a total of 14,833 unique women (G0 mothers) enrolled in ALSPAC as of September 2021 [10]. 12,113 G0 partners have been in contact with the study at some point over the last 30 years by providing data and/or formally enrolling from 2010. 3,807 G0 partners are currently enrolled [11].

## Process

Throughout the life of the cohort, the primary method of data collection has been via self-completion questionnaires. These cover a wide range of topics and have been administered roughly annually over recent years to both generations. As part of questionnaires planned for distribution in the winter of 2020 to both generations (when G1 were around 27 years of age), we chose to test the value of opting in versus opting out.

All G0 and G1 participants with an active email address were sent an invitation to complete the online questionnaire. Participants were not contacted if our administrative database record indicated that they were deceased, had withdrawn from the study, had declined further contact, had declined to complete questionnaires or we did not have up to date contact details for them.

Non-responders were sent a reminder email to complete the online questionnaire. As part of our general questionnaire reminder strategy, a paper version of the questionnaire was then sent to those who had still not complete online or those participants for whom we did not have a valid email address. Respondents were randomised to either opt-out or opt-in with respect to receiving a £10 voucher as a thank you for completing the questionnaire.

At the end of the questionnaire, respondents were reminded that they could receive a voucher as a thankyou for returning a partly or fully completed questionnaire. Those respondents in the opt-out group were asked "If you don't wish to receive your thank you voucher, please, check/cross this box" for the paper/online versions respectively. This compared to the text provided to those in the opt-in group: "If you would like to receive a thank you voucher for completing your questionnaire, please check/cross this box." Given the change in process, those who were randomised to the opt-in process were given the following warning in their invitations: "This year we have changed the way in which we process the vouchers sent out for completing your questionnaire. If you would like to receive a thank-you voucher, please make sure that you check the box at the end of the questionnaire." Vouchers were received if respondents in the opt-in group ticked the box, or if respondents in the opt-out group did not tick the box. We present the results in terms of respondents willing to receive their voucher.

Please note, the study website contains details of all the data that is available through a fully searchable data dictionary and variable search tool. The online version of the questionnaire was developed and deployed using REDCap (Research Electronic Data CAPture tools [12]; a secure web application for building and managing online data collection exercises, hosted at the University of Bristol. Paper questionnaires were designed, scanned and verified using Teleform data capture software.

The questionnaire for G0 was open between 23rd January and 31st October 2020 and for G1 between 15th November 2019 and 31st July 2020.

## Ethics approval and consent to participate

Ethical approval for the study was obtained from the ALSPAC Ethics and Law Committee and the Local Research Ethics Committees. Informed consent for the use of data collected via questionnaires was obtained from all participants following the recommendations of the ALSPAC Ethics and Law Committee at the time. The submission of a completed questionnaire, either on paper or online, was considered to be written consent from participants to use their data for research purposes. Study participants have the right to withdraw their consent for elements of the study or from the study entirely at any time. Full details of the ALSPAC consent procedures are available on the study website (http://www.bristol.ac.uk/alspac/researchers/research-ethics/).

## Baseline characteristics

We considered a number of baseline characteristics including mode of questionnaire completion (online or paper) and participant type (G0 mother, G0 partner, G1 male or female), highest level of education (< O level, O level or > O level, where O levels are compulsory exams taken at the age of 16 years in the UK) and Ethnicity (white, other than white).

## Statistical analyses

Baseline characteristics (education, ethnicity and participant type) of the eligible participants were compared to determine whether randomisation was balanced. We explored whether different groups of eligible participants were more or less likely to complete the questionnaire in the first place, according to whether they were randomised to opt-in or to opt-out using chi-squared tests to examine any differences. Next we compared differences in baseline characteristics and participant type (G0 or G1, male or female) according to whether respondents received a voucher or not. Logistic regressions were used to investigate the characteristics associated with receiving or not receiving a voucher. Unadjusted odds ratios (ORs) and 95% confidence intervals (CIs) were calculated for each characteristic. Adjusted ORs and 95% CIs were calculated, mutually adjusting for all other characteristics. We calculated the total cost of incentives provided in the opt-in and opt-out groups according to respondent's baseline characteristics to determine whether there were any differences in savings when moving to opt-in versus opt-out. Here, we employed chi-squared tests to compare the actual savings with the expected savings, assuming that there were no differences between different groups. Finally, we present the proportion of responders opting in to receive their incentive, who completed questionnaires that were sent out after the COVID-19 pandemic and estimated the savings that were made as a result of moving from an opt-out to an opt-in approach.

## Results

### Response rate

In total, 20,532 eligible participants were invited to complete a questionnaire with 10,737 completing at least one section (response rate of 52.3%; i.e. the number of respondents divided by the number of eligible participants). The questionnaires were completed between January and October 2020 by G0 and between November 2019 and July 2020 by G1.

Overall, 49.7% of all eligible participants were randomised to the opt-in option with 50.3% randomised to the opt-out option. Randomisation was balanced according to eligible participant type, education level and ethnicity (all p>0.05; Table 1).

There were no differences in the proportions of respondents completing the questionnaire who were in the opt-in or opt-out arms (Table 1). Of those who responded, similar proportions of opt-ins and opt-outs were seen according to respondent type or their baseline characteristics (all p>0.10; Table 2).

Overall, 17.8% of respondents chose not to receive their thank you voucher (Table 3). A greater proportion of respondents received a voucher if they were in the opt-out arm (90.7%, compared to 73.7% of those who had to opt-in; unadjusted OR 3.49 (95% CI: 3.12, 3.89).

A greater proportion of G0 mothers than G0 partners received their voucher (77.5% vs 70.2%; Table 3). However, a greater proportion of G1 respondents received their voucher compared to their parents. A slightly higher proportion of G1 females received their voucher compared to G1 males (92.4% vs 89.4%). In terms of education, the lowest proportion of respondents receiving their voucher were in highest category of attainment compared to the

**Table 1. Balance of randomisation according to baseline characteristics and mode of questionnaire completion (n = 20,532 eligible participants who were sent the questionnaire to complete).**

|  | Randomised to opt-in (n = 10,226) | Randomised to opt-out (n = 10,306) |
|---|---|---|
| **Questionnaire Completed** | | |
| No (n = 9795) | 50.0% (n = 4894) | 50.0% (n = 4901) |
| Yes (n = 10,737) | 49.7% (n = 5332) | 50.3% (n = 5405) |
| $X^2 = 0.19$ (p = 0.663) | | |
| **Participant type** | | |
| G0 mother (n = 9391) | 49.8% (n = 4681) | 50.2% (n = 4710) |
| G0 partner (n = 1977) | 50.2% (n = 993) | 49.8% (n = 984) |
| G1 female (n = 4925) | 49.6% (n = 2443) | 50.4% (n = 2482) |
| G1 male (n = 4239) | 49.8% (n = 2109) | 50.2% (n = 2130) |
| $X^2 = 0.23$ (p = 0.972) | | |
| **Education** [a] | | |
| < O level (n = 4339) | 51.0% (n = 2215) | 49.0% (n = 2124) |
| O level (n = 6072) | 50.1% (n = 3043) | 49.9% (n = 3029) |
| > O level (n = 7790) | 49.2% (n = 3833) | 50.8% (n = 3957) |
| $X^2 = 3.89$ (p = 0.143) | | |
| **Ethnicity** | | |
| White (n = 17817) | 49.7% (n = 8849) | 50.3% (n = 8968) |
| Other than white (n = 572) | 50.7% (n = 290) | 49.3% (n = 282) |
| $X^2 = 0.24$ (p = 0.627) | | |

[a] O levels are compulsory exams taken at the age of 16 years in the UK

**Table 2. Participants completing in each group according to baseline characteristics and mode of questionnaire completion (n = 10,737 respondents completing the questionnaire).**

| | Randomised to opt-in (n = 5332) | Randomised to opt-out (n = 5405) |
|---|---|---|
| **Mode of completion** | | |
| Online (n = 8330) | 50.7% (n = 4220) | 49.3% (n = 4110) |
| Paper (n = 2407) | 49.3% (n = 1185) | 50.7% (n = 1222) |
| $X^2$ = 1.53 (p = 0.217) | | |
| **Participant type** | | |
| G0 mother (n = 4828) | 50.1% (n = 2418) | 49.9% (n = 2410) |
| G0 partner (n = 1463) | 49.4% (n = 722) | 50.6% (n = 741) |
| G1 female (n = 2951) | 49.7% (n = 1467) | 50.3% (n = 1484) |
| G1 male (n = 1495) | 48.5% (n = 725) | 51.5% (n = 770) |
| $X^2$ = 1.22 (p = 0.749) | | |
| **Education** [a] | | |
| < O level (n = 1645) | 51.9% (n = 853) | 48.1% (n = 792) |
| O level (n = 3176) | 49.9% (n = 1586) | 50.1% (n = 1590) |
| > O level (n = 5005) | 48.4% (n = 2421) | 51.6% (n = 2584) |
| $X^2$ = 5.43 (p = 0.092) | | |
| **Ethnicity** | | |
| White (n = 9660) | 49.3% (n = 4762) | 50.7% (n = 4898) |
| Other than white (n = 257) | 54.1% (n = 139) | 45.9% (n = 118) |
| $X^2$ = 2.30 (p = 0.130) | | |

[a] O levels are compulsory exams taken at the age of 16 years in the UK

other two groups (79.5% vs 84.0% and 84.5%). A greater proportion of respondents of other than white ethnicity received their voucher (87.2%) compared to whites (81.7%).

After adjustment for all other characteristics, the OR for receiving a voucher increased from 3.49 (95% CI: 3.12, 3.89) to 3.94 (95% CI: 3.49, 4.45) for those in the opt-out group compared to the opt-in group. All other characteristics remained strongly associated with voucher receipt after adjustment (Table 3).

It can be seen in Table 4 presents that differential cost savings were made in the opt-in group versus the opt-out group. Overall the total cost of incentives was £9,740 in the opt-in group compared to the opt-out group. Observed differences were larger than expected in those respondents completing their questionnaires on paper, in the G0 respondents, in the highest educational attainment group and in respondents of white ethnicity (all p<0.001).

## Discussion

By undertaking a nested randomised trial within an existing longitudinal population study, we have shown that respondents are less likely to choose to receive a thankyou for their time in the form of a £10 shopping voucher if they are asked to opt-in compared to if they are asked to opt-out. Larger reductions in the observed cost of incentives were seen in the older generation, those of who higher education, completing on paper and of white ethnicity,

There is already convincing evidence to show that providing monetary incentives improve response rates to randomised controlled trials [2] and retention rates in cohort studies [3], with other birth cohort studies in the UK considering their use to maintain participation rates [13]. However, to our knowledge, this is the first study to formally test the effect of offering an

**Table 3. Proportions of respondents (with odds ratios and 95% confidence intervals) who received a voucher and those who did not according to baseline characteristics (n = 10,737 who completed the questionnaire).**

| | Did not receive voucher n = 1905 [17.8%] | Received voucher n = 8832 [82.2%] | Unadjusted OR (95% CI) (n = 10737) | Adjusted* OR (95% CI) (n = 9720) |
|---|---|---|---|---|
| **Randomisation** | | | | |
| Opt-in (n = 5332) | 26.3% (n = 1403) | 73.7% (n = 3929) | 1.00 | 1.00 |
| Opt-out (n = 5405) | 9.3% (n = 502) | 90.7% (n = 4903) | 3.49 (3.12, 3.89) | 3.94 (3.49, 4.45) |
| $X^2$ = 500.6 (p<0.001) | | | | |
| **Mode of completion** | | | | |
| Online (n = 8330) | 12.9% (n = 1074) | 87.1% (n = 7256) | 1.00 | 1.00 |
| Paper (n = 2407) | 34.5% (n = 831) | 65.5% (n = 1576) | 3.56 (3.21, 3.96) | 3.80 (3.37, 4.29) |
| $X^2$ = 598.7 (p<0.0001) | | | | |
| **Participant type** | | | | |
| G0 mother (n = 4828) | 22.5% (n = 1088) | 77.5% (n = 3740) | 1.00 | 1.00 |
| G0 partner (n = 1463) | 29.8% (n = 436) | 70.2% (n = 1027) | 0.69 (0.68, 0.78) | 0.66 (0.57, 0.76) |
| G1 female (n = 2951) | 7.6% (n = 223) | 92.4% (n = 2728) | 3.56 (3.06, 4.14) | 3.25 (2.75, 3.84) |
| G1 male (n = 1495) | 10.6% (n = 158) | 89.4% (n = 1337) | 2.46 (2.06, 2.94) | 2.36 (1.94, 2.86) |
| $X^2$ = 484.3 (p<0.0001) | | | | |
| **Education** [a] | | | | |
| < O level (n = 1645) | 16.0% (n = 263) | 84.0% (n = 1382) | 1.35 (1.17, 1.57) | 1.61 (1.36, 1.89) |
| O level (n = 3176) | 15.5% (n = 493) | 84.5% (n = 2683) | 1.40 (1.25, 1.58) | 1.42 (1.25, 1.62) |
| > O level (n = 5005) | 20.5% (n = 1025) | 79.5% (n = 3980) | 1.00 | 1.00 |
| $X^2$ = 38.3 (p<0.0001) | | | | |
| **Ethnicity** | | | | |
| White (n = 9660) | 18.3% (n = 1762) | 81.7% (n = 7897) | 1.00 | 1.00 |
| Other than white (n = 257) | 12.8% (n = 33) | 87.2% (n = 224) | 1.52 (1.05, 2.19) | 1.62 (1.08, 2.42) |
| $X^2$ = 4.94 (p = 0.026) | | | | |

*Adjusted for all other variables in the table.

[a] O levels are compulsory exams taken at the age of 16 years in the UK.

opt-in process versus an opt-out process for receiving those incentives. This is important for ongoing studies which routinely offer incentives and offers a potential way to reduce expenditure when administering data collection activities. Whilst these savings are likely to be small over time this could built up into a significant amount. For example, in ALSPAC, if we continue to provide an annual questionnaire, we could save almost £200,000 (Table 5) over the next 10 years by switching to an opt-in rather than an opt-out strategy. We will evaluate the ongoing impact of opt-in over time when we have completed a number of additional data sweeps. It is important to note that financial incentives are one of a range of engagement techniques and here we have not explored their use for re-engaging with participants that might be lost to the study.

Strengths of this study include the longitudinal nature of the wider study and its size. ALSPAC has a core set of engaged participants and over 10,000 responded to the questionnaires that were randomised. We included two generations of participants enabling evidence to be gathered in two distinct age groups and we were able to investigate associations with baseline characteristics. Limitations include the possible timing of these questionnaires; although they were sent out well before the COVID-19 pandemic, our reminder process did crossover into the first lockdown in the UK. G0. However, 94% of G1 respondents and 82% of G0 partners completed their questionnaires before lockdown. 60% of G0 mothers completed

**Table 4. Cost of incentives provided in the opt-in and opt-out groups according to respondent's baseline characteristics (n = 8,832 who completed the questionnaire and opted to receive the voucher).**

| | Opt-in group n = 3929 | Opt-out group n = 4903 | Absolute Difference[b] | Expected difference[c] | X² (p) [d] |
|---|---|---|---|---|---|
| Overall Cost of incentives | £39,290 | £49,030 | £9,740 | | |
| **Mode of completion** | | | | | |
| Online (82.2%) | £34,900 | £37,660 | £2,700 | £8,006 | |
| Paper (17.8%) | £4,390 | £11,370 | £6,980 | £1,734 | 580 (p<0.001) |
| **Participant type** | | | | | |
| G0 mother (42.3%) | £16,190 | £21,210 | £5,020 | £4,120 | 167 (p<0.001) |
| G0 partner (11.6%) | £4,300 | £5,970 | £1,670 | £1,130 | 121 (p<0.001) |
| G1 female (30.9%) | £12,740 | £14,540 | £1,800 | £3,010 | 404 (p<0.001) |
| G1 male (15.2%) | £6,060 | £7,310 | £1,250 | £1,480 | 22.5 (p<0.001) |
| **Education [a]** | | | | | |
| < O level (17.2%) | £6,300 | £7,520 | £1,220 | £1,605 | 61.8 (p<0.001) |
| O level (33.3%) | £12,070 | £14,760 | £2,690 | £3,107 | 43.5 (p<0.001) |
| > O level (49.5%) | £17,190 | £22,610 | £5,420 | £4,618 | 138 (p<0.001) |
| **Ethnicity** | | | | | |
| White (97.2%) | £34,720 | £44,250 | £9,530 | £9,302 | |
| Other than white (2.8%) | £1,110 | £1,130 | £20 | £268 | 216.3 (p<0.001) |

[a] O levels are compulsory exams taken at the age of 16 years in the UK.

[b] Opt-out costs minus opt-in costs.

[c] Calculated assuming no difference according to baseline categories.

[d] For variables with more than 2 categories comparing to all other categories combined.

before the pandemic, however this group had the lowest proportion of respondents receiving vouchers and it is unlikely that the lockdown will have affected their choice to opt-in or opt-out. We have not seen any evidence that changing our protocol to opt-in has affected overall response rates: Table 5 shows that in both generations, response rates were actually higher in the subsequent questionnaire sweep but are then subject to what we would expect given natural attrition. A further limitation is that our results may not be generalisable to a) other types of studies such as cross-sectional studies, household panels or b) those using different monetary structures, types of incentives, such as cash or amounts (e.g. rewards >£10). Finally, there

**Table 5. Response rates and proportion of respondents opting in to receive incentives post-pandemic.**

| Date of questionnaire | G0 | | G1 | |
|---|---|---|---|---|
| | Response rate | % receiving voucher | Response rate | % receiving voucher |
| *2019–20 for comparison* | *55.3%* | *72.3%* | *48.5%* | *85.8%* |
| Mid-2023 | Not done | Not done | 45.6% | 89.2% |
| Early 2023 | 53.0% | 75.6% | 44.1% | 85.0% |
| Mid-2022 | 52.6% | 68.0% | 45.0% | 88.0% |
| Early 2022 | 55.9% | 80.4% | 47.0% | 92.0% |
| 2019/2020 | Not done | Not done | 50.2% | 73.7% |
| Average | | 75% | | 88% |
| **Potential saving per Q** | Assuming 6,000 responses | | Assuming 4,000 responses | |
| | **£15,000** | | **£4,800** | |

is potential that the differences in wording to describe the voucher used in the opt-in ("a thankyou voucher for completing your questionnaire") versus the opt-out ("your thank you voucher") may have affected whether respondents chose to receive their vouchers. Given participants have been with the study for many years and would be used to receiving a voucher as a thank you, it is unlikely that a change in wording would have an effect but it cannot be ruled out.

## Conclusions

The introduction of an opt-in option for receiving a financial incentive has reduced the number of questionnaire respondents who received a voucher in both generations of the ALSPAC study compared to an opt-out option. We recommend that studies which rely on volunteers or members of the public undertaking acts of altruism, but may offer financial incentives may want to consider changing their protocols to opt-in rather than opt-out in order to make small savings.

## Acknowledgments

We are extremely grateful to all the families who took part in this study, the midwives for their help in recruiting them, and the whole ALSPAC team, which includes interviewers, computer and laboratory technicians, clerical workers, research scientists, volunteers, managers, receptionists and nurses.

## Author Contributions

**Conceptualization:** Kate Northstone.

**Data curation:** Kate Northstone.

**Formal analysis:** Kate Northstone.

**Methodology:** Kate Northstone, Claire Bowring.

**Writing – original draft:** Kate Northstone.

**Writing – review & editing:** Claire Bowring.

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
