## [Decision Letter · Decision Letter 0]

17 Oct 2024

PONE-D-24-31090Does opting in or out affect the take up of incentives in a long running population-based cohortstudy: A nested randomised trial in ALSPACPLOS ONE

Dear Dr. Northstone,

Thank you for submitting your manuscript to PLOS ONE. After careful consideration, we feel that it has merit but does not fully meet PLOS ONE’s publication criteria as it currently stands. Therefore, we invite you to submit a revised version of the manuscript that addresses the points raised during the review process.

We look forward to receiving your revised manuscript.

Kind regards,

Adnan Ahmad Khan

Academic Editor

PLOS ONE

Journal Requirements: When submitting your revision, we need you to address these additional requirements. 1. Please ensure that your manuscript meets PLOS ONE's style requirements, including those for file naming. The PLOS ONE style templates can be found at https://journals.plos.org/plosone/s/file?id=wjVg/PLOSOne_formatting_sample_main_body.pdf and https://journals.plos.org/plosone/s/file?id=ba62/PLOSOne_formatting_sample_title_authors_affiliations.pdf 2. Please ensure that you have specified a) Did participants provide their written or verbal informed consent to participate in this study?b) If consent was verbal, please explain i) why written consent was not obtained, ii) how you documented participant consent, and iii) whether the ethics committees/IRB approved this consent procedure." In consent please state in Ethics Method section and manuscript if it is written or verbal. If consent was verbal, please explain a) why written consent was not obtained, b) how you documented participant consent, and c) whether the ethics committees/IRB approved this consent procedure. 3. We note that the grant information you provided in the ‘Funding Information’ and ‘Financial Disclosure’ sections do not match.  When you resubmit, please ensure that you provide the correct grant numbers for the awards you received for your study in the ‘Funding Information’ section. 4. Thank you for stating the following financial disclosure: "The UK Medical Research Council and Wellcome (Grant ref: 217065/Z/19/Z) and the University of Bristol provide core support for ALSPAC. This publication is the work of the authors and KN will serve as guarantor for the contents of this paper. A comprehensive list of grants funding is available on the ALSPAC website (http://www.bristol.ac.uk/alspac/external/documents/grant-acknowledgements.pdf)."  Please state what role the funders took in the study.  If the funders had no role, please state: ""The funders had no role in study design, data collection and analysis, decision to publish, or preparation of the manuscript."" If this statement is not correct you must amend it as needed. Please include this amended Role of Funder statement in your cover letter; we will change the online submission form on your behalf. 5. Please review your reference list to ensure that it is complete and correct. If you have cited papers that have been retracted, please include the rationale for doing so in the manuscript text, or remove these references and replace them with relevant current references. Any changes to the reference list should be mentioned in the rebuttal letter that accompanies your revised manuscript. If you need to cite a retracted article, indicate the article’s retracted status in the References list and also include a citation and full reference for the retraction notice.

**Additional Editor Comments:**

Thank you for submitting the study for review. While we have had mixed reviews, I tend to agree with Reviewer 2 that the study would require minor revisions before being accepted for publication. I am attaching the two reviews below for guidance. In addition, please review the discussion. I would feel that the primary learning from your paper is to inform about opt in v opt out behaviors in situations where people are volunteering or doing altruistic acts. This may be elaborated upon the the discussion.

Reviewer 1

Comments to the Author

1. Is the manuscript technically sound, and do the data support the conclusions?

The manuscript must describe a technically sound piece of scientific research with data that supports the conclusions. Experiments must have been conducted rigorously, with appropriate controls, replication, and sample sizes. The conclusions must be drawn appropriately based on the data presented. Partly

2. Has the statistical analysis been performed appropriately and rigorously? Yes

3. Have the authors made all data underlying the findings in their manuscript fully available?

The PLOS Data policy requires authors to make all data underlying the findings described in their manuscript fully available without restriction, with rare exception (please refer to the Data Availability Statement in the manuscript PDF file). The data should be provided as part of the manuscript or its supporting information, or deposited to a public repository. For example, in addition to summary statistics, the data points behind means, medians and variance measures should be available. If there are restrictions on publicly sharing data—e.g. participant privacy or use of data from a third party—those must be specified. No

4. Is the manuscript presented in an intelligible fashion and written in standard English?

PLOS ONE does not copyedit accepted manuscripts, so the language in submitted articles must be clear, correct, and unambiguous. Any typographical or grammatical errors should be corrected at revision, so please note any specific errors here. Yes

5. Review Comments to the Author

Please use the space provided to explain your answers to the questions above. You may also include additional comments for the author, including concerns about dual publication, research ethics, or publication ethics. (Please upload your review as an attachment if it exceeds 20,000 characters) Referee report for : “Does opting in or out affect the take up of incentives in a long running population based cohort study: A nested randomised trial in ALSPAC”

Summary: This paper uses the Avon Longitudinal Study of Parents and Children to assess whether offering an opt-in or opt-out option for receiving financial incentives when completing questionnaires can result in cost savings. For more than 10 years, this survey has provided a £10 incentive for completing annual questionnaires, automatically given unless participants chose to opt out. For the 2020 questionnaire, eligible parents and their offspring were randomly assigned to either an opt-out or an opt-in option for receiving their £10 vouchers. The authors find that respondents are less likely to claim the £10 shopping voucher when they are required to opt in, compared to when they are required to opt out. They find no difference in response rates between those randomized to the opt-in group and those in the opt-out group. The authors recommend implementing an opt-in system rather than an opt-out system, as it would enable cost savings without negatively affecting the overall response rate.

This paper addresses an ongoing topic regarding the need to use incentives in various types of studies to increase take-up, response rates, RCT participation, etc. The authors focus on a very specific case: within a longitudinal study where a payment scheme was introduced several years after the survey's start, a new option arises—the possibility to opt in for payment. The results is definitely interesting but cannot be easily generalized to other types of studies. Moreover, the design is not perfectly clean: in the paper's final section, when listing the study's limitations, the authors mention a wording difference between the two groups.

Major comments

1) The authors should acknowledge the difference between being incentivized with a voucher for completing a questionnaire and being incentivized with a voucher based on an action (for instance, getting screened) or the outcome of a behavior (e.g. losing weight). I believe the mechanisms triggered by these incentives are not fully comparable. The authors reference the literature on different types of incentives and studies (RCTs, surveys, etc.), which might be misleading for the reader.

2) The structure of the incentives: what the authors tested here is a specific incentive structure, which is a one-time fixed payment. The results cannot be generalized to other types of incentive structures or amounts.

3) Page 3, l. 53: The authors claim that across both generations, they anticipate upwards of 10,000 completed questionnaires. Could the authors clarify the basis for this figure? I would expect the actual number of completed questionnaires up to 2020 to be available, rather than an estimated number. Instead of this assertion, could the authors provide the actual expenditure incurred by this survey? Or maybe I misunderstood the purpose of this sentence.

4) Could it be that respondents prefer cash over a voucher, which is why the voucher is sometimes not used? Could offering cash be a more effective way to incentivize responses? I would be interested in the authors’ opinion on this.

5) I understand that the opt-out option was only recently introduced (p. 3, l. 63). Could the authors explain the reason for this decision? What has been the evolution of the opt-out rate over time?

6) The content in the "Method" section does not actually describe the methods but rather outlines the objectives of the study. This section should be revised. For easier reading, figures and tables should be integrated into the text rather than placed at the end of the paper. Overall, the paper does not seem to follow all the journal's requested guidelines.

7) Is it feasible to estimate the probability of opting in, conditional on having used the vouchers in the past? This would require an ID number linking the respondents to the vouchers, which might not be available.

8) Would the respondents receive the voucher if they completed at least part of the questionnaire or only if they completed the entire questionnaire? Could the authors clarify this?

9) 9) I would be cautious with the results: participants in the opt-in group were specifically informed about a change compared to previous years. In fact, this constitutes a treatment in itself (framing effect), making it difficult to determine whether the observed effect is due to the participants having received a voucher every year for X years and feeling guilty, leading them to prefer to opt out, or if they would have made the same decision based on a one-time offer.

10) Of course, the aim of longitudinal studies is to repeat observations over time for the same respondents. Could the authors indicate if the response rate differs according to the treatment group (opt-in or opt-out) and based on previous responses in earlier years?

11) The mode of completion seems to matter, could the author provide an explanation?

Minor:

1) Page 3, l.51: The authors use the term “high street shopping voucher.” While this may be a common term in the UK, it would be helpful to add a footnote explaining the types of shops where these vouchers can be redeemed.

2) Page 3 l.60: Could the authors provide the actual drop rate instead of “>5%”? Is this drop compared to the previous year or overall? Did it increase again the following year, and if so, by how much?

3) Page 6 l.124: The author could rewrite: “we present the results in terms of respondents willing to receive their voucher.”

4) Could the authors provide more information earlier in the paper about how the questionnaires were completed (e.g., by phone, online)? Has the method of completing the questionnaire changed over time?

5) Do the authors have access to more detailed characteristics, such as income, more levels of education, or living area (rural vs. urban), that could be included in the model?

6. PLOS authors have the option to publish the peer review history of their article (what does this mean?). If published, this will include your full peer review and any attached files.

Do you want your identity to be public for this peer review? For information about this choice, including consent withdrawal, please see our Privacy Policy.

No

Confidential to Editor

1. Do you have any potential or perceived competing interests that may influence your review? Please review our Competing Interests policy and declare any potential interests that you feel the Editor should be aware of when considering your review. If you have no competing interests, please write "I have no competing interests." I have no competing interests

2. Did you receive any assistance in preparing this review (e.g. from a post-doc or graduate student)? If yes, please include their name below.

3. If accepted, do you think this submission should be highlighted on the PLOS ONE website? PLOS ONE does not evaluate manuscripts based on perceived significance or readership. We aim to provide tools for readers to filter and evaluate our publications. (optional)

Do you want to get recognition for this review on a Web of Science researcher profile?

If you opt in, your Web of Science profile will automatically be updated to show a verified record of this review in full compliance with the journal’s review policy. If you don’t have a Web of Science profile, you will be prompted to create a free account. Yes

Reviewer 2

Reviewer Recommendation Term: Minor Revision

Rate Review: 0

Custom Review Question(s): Response

Comments to the Author

1. Is the manuscript technically sound, and do the data support the conclusions?

The manuscript must describe a technically sound piece of scientific research with data that supports the conclusions. Experiments must have been conducted rigorously, with appropriate controls, replication, and sample sizes. The conclusions must be drawn appropriately based on the data presented. Yes

2. Has the statistical analysis been performed appropriately and rigorously? Yes

3. Have the authors made all data underlying the findings in their manuscript fully available?

The PLOS Data policy requires authors to make all data underlying the findings described in their manuscript fully available without restriction, with rare exception (please refer to the Data Availability Statement in the manuscript PDF file). The data should be provided as part of the manuscript or its supporting information, or deposited to a public repository. For example, in addition to summary statistics, the data points behind means, medians and variance measures should be available. If there are restrictions on publicly sharing data—e.g. participant privacy or use of data from a third party—those must be specified. Yes

4. Is the manuscript presented in an intelligible fashion and written in standard English?

PLOS ONE does not copyedit accepted manuscripts, so the language in submitted articles must be clear, correct, and unambiguous. Any typographical or grammatical errors should be corrected at revision, so please note any specific errors here. Yes

5. Review Comments to the Author

Please use the space provided to explain your answers to the questions above. You may also include additional comments for the author, including concerns about dual publication, research ethics, or publication ethics. (Please upload your review as an attachment if it exceeds 20,000 characters) The paper is well developed and consists of good flow of writing. Its needs some revision in the methodology sections and conclusion section. How do you conclude your research as "We recommend similar studies consider this option if they want to introduce some cost savings without harming overall response rates." Is your research does not have any specific findings that drive to conclusion?

Please be specific.

6. PLOS authors have the option to publish the peer review history of their article (what does this mean?). If published, this will include your full peer review and any attached files.

Do you want your identity to be public for this peer review? For information about this choice, including consent withdrawal, please see our Privacy Policy.

No

Confidential to Editor

1. Do you have any potential or perceived competing interests that may influence your review? Please review our Competing Interests policy and declare any potential interests that you feel the Editor should be aware of when considering your review. If you have no competing interests, please write "I have no competing interests." The paper is well developed and consists of good flow of writing. Its needs some revision in the methodology sections and conclusion section. How do you conclude your research as "We recommend similar studies consider this option if they want to introduce some cost savings without harming overall response rates." Is your research does not have any specific findings that drive to conclusion?

It needs more clarification in the research methods to support to the theoretical framework of the research.

2. Did you receive any assistance in preparing this review (e.g. from a post-doc or graduate student)? If yes, please include their name below. N/A

3. If accepted, do you think this submission should be highlighted on the PLOS ONE website? PLOS ONE does not evaluate manuscripts based on perceived significance or readership. We aim to provide tools for readers to filter and evaluate our publications. (optional) Yes, on a broad subject area page (e.g. Biology, Earth Sciences)

Do you want to get recognition for this review on a Web of Science researcher profile?

If you opt in, your Web of Science profile will automatically be updated to show a verified record of this review in full compliance with the journal’s review policy. If you don’t have a Web of Science profile, you will be prompted to create a free account. Yes

Reviewers' comments:

Reviewer's Responses to Questions

**Comments to the Author**

1. Is the manuscript technically sound, and do the data support the conclusions?

Reviewer #1: Partly

Reviewer #2: Yes

2. Has the statistical analysis been performed appropriately and rigorously? 

Reviewer #1: Yes

Reviewer #2: Yes

3. Have the authors made all data underlying the findings in their manuscript fully available?

Reviewer #1: No

Reviewer #2: Yes

4. Is the manuscript presented in an intelligible fashion and written in standard English?

Reviewer #1: Yes

Reviewer #2: Yes

5. Review Comments to the Author

Reviewer #1: Referee report for : “Does opting in or out affect the take up of incentives in a long running population based cohort study: A nested randomised trial in ALSPAC”

Summary: This paper uses the Avon Longitudinal Study of Parents and Children to assess whether offering an opt-in or opt-out option for receiving financial incentives when completing questionnaires can result in cost savings. For more than 10 years, this survey has provided a £10 incentive for completing annual questionnaires, automatically given unless participants chose to opt out. For the 2020 questionnaire, eligible parents and their offspring were randomly assigned to either an opt-out or an opt-in option for receiving their £10 vouchers. The authors find that respondents are less likely to claim the £10 shopping voucher when they are required to opt in, compared to when they are required to opt out. They find no difference in response rates between those randomized to the opt-in group and those in the opt-out group. The authors recommend implementing an opt-in system rather than an opt-out system, as it would enable cost savings without negatively affecting the overall response rate.

This paper addresses an ongoing topic regarding the need to use incentives in various types of studies to increase take-up, response rates, RCT participation, etc. The authors focus on a very specific case: within a longitudinal study where a payment scheme was introduced several years after the survey's start, a new option arises—the possibility to opt in for payment. The results is definitely interesting but cannot be easily generalized to other types of studies. Moreover, the design is not perfectly clean: in the paper's final section, when listing the study's limitations, the authors mention a wording difference between the two groups.

Major comments

1) The authors should acknowledge the difference between being incentivized with a voucher for completing a questionnaire and being incentivized with a voucher based on an action (for instance, getting screened) or the outcome of a behavior (e.g. losing weight). I believe the mechanisms triggered by these incentives are not fully comparable. The authors reference the literature on different types of incentives and studies (RCTs, surveys, etc.), which might be misleading for the reader.

2) The structure of the incentives: what the authors tested here is a specific incentive structure, which is a one-time fixed payment. The results cannot be generalized to other types of incentive structures or amounts.

3) Page 3, l. 53: The authors claim that across both generations, they anticipate upwards of 10,000 completed questionnaires. Could the authors clarify the basis for this figure? I would expect the actual number of completed questionnaires up to 2020 to be available, rather than an estimated number. Instead of this assertion, could the authors provide the actual expenditure incurred by this survey? Or maybe I misunderstood the purpose of this sentence.

4) Could it be that respondents prefer cash over a voucher, which is why the voucher is sometimes not used? Could offering cash be a more effective way to incentivize responses? I would be interested in the authors’ opinion on this.

5) I understand that the opt-out option was only recently introduced (p. 3, l. 63). Could the authors explain the reason for this decision? What has been the evolution of the opt-out rate over time?

6) The content in the "Method" section does not actually describe the methods but rather outlines the objectives of the study. This section should be revised. For easier reading, figures and tables should be integrated into the text rather than placed at the end of the paper. Overall, the paper does not seem to follow all the journal's requested guidelines.

7) Is it feasible to estimate the probability of opting in, conditional on having used the vouchers in the past? This would require an ID number linking the respondents to the vouchers, which might not be available.

8) Would the respondents receive the voucher if they completed at least part of the questionnaire or only if they completed the entire questionnaire? Could the authors clarify this?

9) 9) I would be cautious with the results: participants in the opt-in group were specifically informed about a change compared to previous years. In fact, this constitutes a treatment in itself (framing effect), making it difficult to determine whether the observed effect is due to the participants having received a voucher every year for X years and feeling guilty, leading them to prefer to opt out, or if they would have made the same decision based on a one-time offer.

10) Of course, the aim of longitudinal studies is to repeat observations over time for the same respondents. Could the authors indicate if the response rate differs according to the treatment group (opt-in or opt-out) and based on previous responses in earlier years?

11) The mode of completion seems to matter, could the author provide an explanation?

Minor:

1) Page 3, l.51: The authors use the term “high street shopping voucher.” While this may be a common term in the UK, it would be helpful to add a footnote explaining the types of shops where these vouchers can be redeemed.

2) Page 3 l.60: Could the authors provide the actual drop rate instead of “>5%”? Is this drop compared to the previous year or overall? Did it increase again the following year, and if so, by how much?

3) Page 6 l.124: The author could rewrite: “we present the results in terms of respondents willing to receive their voucher.”

4) Could the authors provide more information earlier in the paper about how the questionnaires were completed (e.g., by phone, online)? Has the method of completing the questionnaire changed over time?

5) Do the authors have access to more detailed characteristics, such as income, more levels of education, or living area (rural vs. urban), that could be included in the model?

Reviewer #2: The paper is well developed and consists of good flow of writing. Its needs some revision in the methodology sections and conclusion section. How do you conclude your research as "We recommend similar studies consider this option if they want to introduce some cost savings without harming overall response rates." Is your research does not have any specific findings that drive to conclusion?

Please be specific.

6. PLOS authors have the option to publish the peer review history of their article (what does this mean?). If published, this will include your full peer review and any attached files.

Reviewer #1: No

Reviewer #2: No

---

## [Author Response · Author response to Decision Letter 0]

4 Dec 2024

Editorial comments:

- Thankyou - we have amended the manuscript where appropriate – this has focussed on the title page, changing headings, double spacing and moving tables to within the main text. Please note we have not used tracked changes but have for all other changes.

2. Please ensure that you have specified a) Did participants provide their written or verbal informed consent to participate in this study?

b) If consent was verbal, please explain i) why written consent was not obtained, ii) how you documented participant consent, and iii) whether the ethics committees/IRB approved this consent procedure."

 In consent please state in Ethics Method section and manuscript if it is written or verbal. If consent was verbal, please explain a) why written consent was not obtained, b) how you documented participant consent, and c) whether the ethics committees/IRB approved this consent procedure.

- We have clarified the consent process (lines 173-175) 

 - Apologies – I will ensure this is correct when resubmitting – the details in the manuscript are correct.

"The UK Medical Research Council and Wellcome (Grant ref: 217065/Z/19/Z) and the University of Bristol provide core support for ALSPAC. This publication is the work of the authors and KN will serve as guarantor for the contents of this paper. A comprehensive list of grants funding is available on the ALSPAC website 

(http://www.bristol.ac.uk/alspac/external/documents/grant-acknowledgements.pdf)."

- Thankyou we have updated the manuscript to this effect. This has also been added to the cover letter as requested.

 - We have checked the reference list and confirm it is complete and correct. We have made two additions to the reference list (new references 4 and 5) but confirm that no citations have been retracted.

Additional Editor Comments:

Thank you for submitting the study for review. While we have had mixed reviews, I tend to agree with Reviewer 2 that the study would require minor revisions before being accepted for publication. I am attaching the two reviews below for guidance. In addition, please review the discussion. I would feel that the primary learning from your paper is to inform about opt in v opt out behaviors in situations where people are volunteering or doing altruistic acts. This may be elaborated upon the the discussion.

- Thankyou for your positive comments. We hope that the changes that we have made in response to both reviewers are acceptable. We have reviewed the discussion as suggested and have adjusted the primary learning as suggested (this has also filtered to the abstract, line 39).

Reviewer #1: 

Summary: This paper uses the Avon Longitudinal Study of Parents and Children to assess whether offering an opt-in or opt-out option for receiving financial incentives when completing questionnaires can result in cost savings. For more than 10 years, this survey has provided a £10 incentive for completing annual questionnaires, automatically given unless participants chose to opt out. For the 2020 questionnaire, eligible parents and their offspring were randomly assigned to either an opt-out or an opt-in option for receiving their £10 vouchers. The authors find that respondents are less likely to claim the £10 shopping voucher when they are required to opt in, compared to when they are required to opt out. They find no difference in response rates between those randomized to the opt-in group and those in the opt-out group. The authors recommend implementing an opt-in system rather than an opt-out system, as it would enable cost savings without negatively affecting the overall response rate.

This paper addresses an ongoing topic regarding the need to use incentives in various types of studies to increase take-up, response rates, RCT participation, etc. The authors focus on a very specific case: within a longitudinal study where a payment scheme was introduced several years after the survey's start, a new option arises—the possibility to opt in for payment. The results is definitely interesting but cannot be easily generalized to other types of studies. Moreover, the design is not perfectly clean: in the paper's final section, when listing the study's limitations, the authors mention a wording difference between the two groups.

Major comments

1) The authors should acknowledge the difference between being incentivized with a voucher for completing a questionnaire and being incentivized with a voucher based on an action (for instance, getting screened) or the outcome of a behavior (e.g. losing weight). I believe the mechanisms triggered by these incentives are not fully comparable. The authors reference the literature on different types of incentives and studies (RCTs, surveys, etc.), which might be misleading for the reader.

- Thank you for your comment. We have added to the background recognising the high level of heterogeneity included in research questions and methodology from the cited systematic review (lines 48-50). We have also added further acknowledgement in our conclusions (line 321; already noted in the abstract) that our recommendations are specific to questionnaire completion.

2) The structure of the incentives: what the authors tested here is a specific incentive structure, which is a one-time fixed payment. The results cannot be generalized to other types of incentive structures or amounts.

- Thankyou for highlighting this – we have clarified this in the discussion (lines 310-312).

3) Page 3, l. 53: The authors claim that across both generations, they anticipate upwards of 10,000 completed questionnaires. Could the authors clarify the basis for this figure? I would expect the actual number of completed questionnaires up to 2020 to be available, rather than an estimated number. Instead of this assertion, could the authors provide the actual expenditure incurred by this survey? Or maybe I misunderstood the purpose of this sentence.

 - Apologies for the confusion here. This statement is based on previous response rates such that for any new round of questionnaires to both G0 and G1 we would expect more than 10,000 responses. This is purely setting the scene for costs. 

4) Could it be that respondents prefer cash over a voucher, which is why the voucher is sometimes not used? Could offering cash be a more effective way to incentivize responses? I would be interested in the authors’ opinion on this.

 - We agree with the reviewer and have discussed the provision of cash at length with both our participants advisory board and our Ethics committee and it has been agreed that this is not possible for a number of reasons For remote data collection activities, it was deemed inappropriate to send cash through the post – this would not be possible for participants where we only have an email address as the most up to date contact point. We could collect bank details and arrange bank transfers but this data is deemed too sensitive to keep on file and the administrative burden would far outweigh any potential financial savings. Shopping vouchers are sent via email (or in the post when explicitly requested or when the participant has completed their questionnaire via post) and the brand we use offer the widest possible usage – including supermarkets which in the current cost of living crisis is felt to the most appropriate.

5) I understand that the opt-out option was only recently introduced (p. 3, l. 63). Could the authors explain the reason for this decision? What has been the evolution of the opt-out rate over time?

 - We have addressed this by expanding on the background (please see lines 78-87). This was primarily an effort to see whether we could save costs. At the time of writing we have yet to evaluate any further data collection sweeps and have added to this effect in the discussion (lines 287-289). 

6) The content in the "Method" section does not actually describe the methods but rather outlines the objectives of the study. This section should be revised. For easier reading, figures and tables should be integrated into the text rather than placed at the end of the paper. Overall, the paper does not seem to follow all the journal's requested guidelines.

 - The subsection entitled “Process”, details the methods that we use. Does the reviewer have a particular area that they feel is missing here? – we would be very happy to amend. We have attended to the journal’s guidelines as noted above in response to the editorial comments.

7) Is it feasible to estimate the probability of opting in, conditional on having used the vouchers in the past? This would require an ID number linking the respondents to the vouchers, which might not be available.

 - Thank you for this useful suggestion. We can indeed link to previous data completion exercises given the nature of this longitudinal study. However, the whole point of changing opt-out to opt-in is to test the “default effect” and so we do not think that conditional probabilities would be helpful here. 

8) Would the respondents receive the voucher if they completed at least part of the questionnaire or only if they completed the entire questionnaire? Could the authors clarify this?

 - Apologies if this was not clear. Participants receive the voucher for part or full completion. We have clarified this in the abstract (lines 23-24), the background section (line 56) and the methods (lines 97 and 144).

9) 9) I would be cautious with the results: participants in the opt-in group were specifically informed about a change compared to previous years. In fact, this constitutes a treatment in itself (framing effect), making it difficult to determine whether the observed effect is due to the participants having received a voucher every year for X years and feeling guilty, leading them to prefer to opt out, or if they would have made the same decision based on a one-time offer.

- We explicitly told participants that the process had changed (see lines 150-153). This was as a result of a steer we received from our ethics committed to ensure that those who wanted the voucher did not miss out as they would expect to receive it based on previous experience of not doing anything to receive the voucher. We therefore feel participants were encouraged to take action to receive the voucher rather than guilt them into not receiving it. The language used was purposefully neutral and received full approval from our ethics committee. 

10) Of course, the aim of longitudinal studies is to repeat observations over time for the same respondents. Could the authors indicate if the response rate differs according to the treatment group (opt-in or opt-out) and based on previous responses in earlier years?

 - It is difficult to untangle the effects on response rates that might be due to the ‘treatment group’ compared to the expected attrition rates one would expect to see after 30 years of regular data collection. However, response rates were not affected by the treatment group as described in Table 1 ( and lines 227-228). They have remained consistent over the last 5 years.

11)) The mode of completion seems to matter, could the author provide an explanation?

 - We have been actively encouraging our participants to complete questionnaires online rather than paper from both cost (postage, printing, staff costs to process) and environmental points of view. However, the participants who still complete on paper are different to those who complete online – they are much more likely to be in the older G0 cohort and to have lower levels of education, we therefore considered it an important factor to consider. 

Minor:

1) Page 3, l.51: The authors use the term “high street shopping voucher.” While this may be a common term in the UK, it would be helpful to add a footnote explaining the types of shops where these vouchers can be redeemed.

- Thankyou for raising this point. We have added detail as requested (lines 57-58).

2) Page 3 l.60: Could the authors provide the actual drop rate instead of “>5%”? Is this drop compared to the previous year or overall? Did it increase again the following year, and if so, by how much?

 - We have been specific as requested (now line 69) and added details on the previous and subsequent questionnaires. 

3) Page 6 l.124: The author could rewrite: “we present the results in terms of respondents willing to receive their voucher.”

- Amended as suggested (now lines 155-156)

4) Could the authors provide more information earlier in the paper about how the questionnaires were completed (e.g., by phone, online)? Has the method of completing the questionnaire changed over time?

- This was described in the original version - now lines 136-138. As we have noted above (response to major point 11) we are moving away from paper to online. 

5) Do the authors have access to more detailed characteristics, such as income, more levels of education, or living area (rural vs. urban-), that could be included in the model?)

- In our experience education provides by far the best SES measure in ALSPAC and tends to trump any other measures. The aim of the study was not to predict who exactly was more likely to opt-in/our or not. We therefore would prefer not to add any further analyses and will be guided by the editor on this. 

Reviewer #2: 

The paper is well developed and consists of good flow of writing. Its needs some revision in the methodology sections and conclusion section. How do you conclude your research as "We recommend similar studies consider this option if they want to introduce some cost savings without harming overall response rates." Is your research does not have any specific findings that drive to conclusion?

Please be specific.

- Thank you for your positive comments. It is not clear where in particular you might like us to revise the methods. However, in line with the editor we have updated the conclusions.

---

## [Editor Report · Decision Letter 1]

13 Dec 2024

Does opting in or out affect the take up of incentives in a long running population-based cohortstudy: A nested randomised trial in ALSPAC

PONE-D-24-31090R1

Dear Dr. Northstone,

We’re pleased to inform you that your manuscript has been judged scientifically suitable for publication and will be formally accepted for publication once it meets all outstanding technical requirements.

Kind regards,

Adnan Ahmad Khan

Academic Editor

PLOS ONE
---

## [Editor Report · Acceptance letter]

17 Jan 2025

PONE-D-24-31090R1 

PLOS ONE

Dear Dr. Northstone, 

I'm pleased to inform you that your manuscript has been deemed suitable for publication in PLOS ONE. Congratulations! Your manuscript is now being handed over to our production team.

Kind regards, 

on behalf of

Dr Adnan Ahmad Khan 

Academic Editor

PLOS ONE
